# Radicalization Processes and Transitional Phases in Female and Male Detainees Residing in Dutch Terrorism Wings

**DOI:** 10.3390/bs13100877

**Published:** 2023-10-23

**Authors:** Gaby Thijssen, Jelle Sijtsema, Stefan Bogaerts, Lys van de Voorde, Erik Masthoff

**Affiliations:** 1Tilburg School of Social and Behavioral Sciences, Tilburg University, 5037 AB Tilburg, The Netherlandss.bogaerts@tilburguniversity.edu (S.B.); e.masthoff@fivoor.nl (E.M.); 2Fivoor Academy of Research, Innovation and Development (FARID), 3170 AA Poortugaal, The Netherlands; 3Institute for Criminal Law and Criminology, University of Leiden, 2311 EZ Leiden, The Netherlands; l.van.de.voorde@law.leidenuniv.nl

**Keywords:** radicalization processes, turning points, deradicalization, violent extremism

## Abstract

Background: Radicalization, violent extremism, and terrorism are risks to societal security. Although research on terrorism-related behaviors is increasing, thorough empirical studies are rare. Methods: This study investigates radicalization processes and transitions in a matched sample of female and male terrorist suspects and convicts (N = 26) residing in Dutch penitentiary terrorism wings. Results: Results show that both men and women often experienced discrimination. A subgroup of women grew up in a stressful family environment and lacked emotional support from their family, whereas the other women did not experience such circumstances. The majority of the study sample was susceptible to connecting with radicalized friends or family members. Interestingly, factors that initially led to radicalization (e.g., a utopian image of the Islamic State) could later turn out to be factors associated with abandoning extremism. Conclusions: In this study, differences in radicalization processes and transitional phases between women and men emerged. Men more often had police contact prior to a terrorism-related offense. Making an effort to right old mistakes seemed important in the radicalization processes of men, whereas women had a stronger desire for emotional support and were more driven by experienced trauma and feelings of loneliness. This study provides input for gender-specific prevention and disengagement interventions.

## 1. Introduction

The large number of terroristic attacks around the world in recent decades renewed interest in radicalization processes that may lead to violent extremism [1]. In this study, radicalization is defined as a process of increasing willingness to accept and act upon the ultimate consequence of a line of thought. This increasing willingness can lead to behavior that deeply hurts other people or affects their freedom, can lead individuals or groups to turn away from society, and can lead to violence [2]. In radicalization processes, ideology serves as a framework for the actions of individuals who become radicalized. Although factors contributing to an individual’s radicalization are complex, radicalization processes are generally influenced by well-known mechanisms (e.g., cognitive constructs and behavioral patterns; [3]. As a result, criminal extremism is not considered to be an inherent psychological characteristic or character trait. Instead, it is acquired through cognitive and behavioral changes that occur within a social context, influenced by environmental factors, such as peers and media [3]. Violent extremism is defined as the ideological motives of a person or group to seriously violate the law to engage in activities that undermine the democratic constitutional state [2]. Criminal (here: violent extremist) behavior is acquired through a similar process as other forms of “work-related behavior” [3]. Violent extremists are expected to adapt to the expectations of their new roles. The extremist ideology offers them values and norms, delineates moral boundaries, and establishes a clear perception of the enemy. In this study, individuals who embrace an ideology that justifies violence and adapt to their new roles (thereby breaking the law) are considered violent extremists. Furthermore, there are different definitions of terrorism [4]. In the current study, terrorism is defined as committing lethal violence based on ideological motives, or causing societal damage to property, with the aim of causing social undermining and destabilization, seriously encouraging the population of a country, or influencing political decision-making [2].

Previous research on violent extremism has focused more on men than women [5]. In this study, we emphasize women’s radicalization processes, since they also have a role in terrorist activities [6]. Attention to both male and female radicalization is necessary to contribute to the prevention of future violent extremism [7,8]. There is evidence that the radicalization process occurs differently for women and therefore may require different prevention and intervention strategies than for men [9]. This study thus investigates why men and women move toward (extremist) violence and what the underlying dynamics of these processes are.

### 1.1. Radicalization, Personal Constructs, and the Personal Construct Theory

Various theories shed light on the factors underlying terrorism and violent extremism. For example, the General Strain Theory [10] states that terrorism often arises when individuals endure unjust collective strains influenced by significantly more powerful entities and the act of terror carries a high magnitude of civilian impact [10]. Meanwhile, the Age-Graded Theory of Informal Social Control suggests that individuals with weakened social ties are more likely to commit violent (extremist) acts [11,12,13]. When those ties are weakened, individuals are more likely to commit crimes [14]. Instead, the Significant Quest Theory emphasizes the activation of a quest for significance triggered by significant loss or potential gain, driving individuals towards violent (extremist) behaviors rooted in commitment to vital ideological values [15]. Furthermore, according to the Social Movement Theory, radicalization is framed as “a set of opinions and beliefs in a population, which represents preferences for changing some elements of the social structure and/or reward distribution of a society” (p. 17, [16]). In general, prior to radicalization, individuals often experience personal and social strain and uncertainty [17], making them susceptible to identification with violent extremists. This identification provides a sense of certainty and may include justifications for violent actions against others [18].

According to the Personal Construct Theory (PCT; [19]), an individual’s cognitive constructs are integral to their identity. Fear arises as soon as individuals find out that their cognitive constructs fail to enable them to predict events. This anxiety can be alleviated through the modification of these constructs [18]. Translated to the process of (de)radicalization, the PCT model integrates existing models (e.g., cognitive psychological models) focused on individual and societal processes and change ([18] see Figure 1). According to the proposed model, radicalization is preceded by the presence of various strains (e.g., family dynamics, rejection, humiliation, or other perceived grievances) that contribute to a sense of insecurity by undermining existing cognitive constructs [10,18]. When cognitive constructs are rapidly invalidated, individuals become deprived of a cognitive structure to anticipate and understand events, resulting in feelings of fear, sadness, rage, or emptiness. The only resource of cognitive reconstruction is to construct the person or event on the opposite end of the construct dimension (e.g., “if the government is not good, it must be bad”). This new cognitive construct may involve an extremely negative perception of the other group [18]. Individuals who adopt this new cognitive construct often restrict their existing social contacts and actively seek connections with violent extremists who share similar constructs [10]. Becoming a member of this new group of violent extremists is termed “identity fusion” [18]. In this, individuals share the same cognitive constructs, view the same out-group negatively, and benefit from social support and a sense of unity within the violent extremist group [10,18]. Some radicalized individuals engage in violent extremist acts to align more closely with their new role, particularly when they perceive themselves as warriors or martyrs, leading to unwavering commitment without considering alternatives [18]. When individuals who see themselves as gentle commit violent acts, they may experience guilt. However, when violence is embedded in their cognitive construct as something worthy or an ultimate life goal, such guilt is mitigated. These radical cognitive constructs persist as long as individuals ignore invalidation. This often propels individuals further into violent extremism, as it provides a feeling of certainty within their self-construct. Seeing themselves as non-violent would conflict with their existing construct. Disengagement from extremism can also be explained within the framework of the PCT model. It may occur when individuals experience major invalidation of their radical cognitive constructs or when alternative cognitive constructs are available and validated by like-minded individuals [18].

Moreover, to understand radicalization trajectories, it is important to consider how individual processes interact with the environment [20]. Indeed, the environment feeds individual constructs, for example through peers and the media. When there is disconnection with the environment, personal constructs can lead to hostility and radicalization. Here, it is important to capture the timing of these events and to highlight that many of these interactions occur differently across human development [21].

### 1.2. Development of Violent Extremist Behavior in Radicalization Processes

Furthermore, research highlights other factors contributing to radicalization processes. These factors include personal or social crises, psychological vulnerability, or rewards [22], and problematic home situations, such as coming from a broken home or being exposed to parental conflict [21,23,24,25]. In addition to personal factors, environmental factors also play a role in radicalization processes. For instance, discrimination, marginalization, and stigmatization are important triggers for radicalization [21] and have been associated with perceived threats to the Muslim community and disconnection from society [4,21]. Furthermore, group norms and group prejudices are often associated with radicalization [22] for example, emphasizing collective identity, strengthening one’s ideology, and looking at society from an ‘us versus them’ perspective [26,27,28]. Previous studies showed that many violent extremists in Europe belong to the second-generation of Muslim immigrants [29,30]. They often seek a balance between their newly adopted practical European lifestyle and the cultural heritage they received from their parents [31,32]. These individuals carry the weight of their ethnic and religious backgrounds while also feeling compelled to navigate between different identities to maintain family connections while integrating into European society [31,32]. Identity uncertainty and identity fusion are associated with quests for meaning in life. Some individuals find answers to these existential questions by engaging in violent extremism [32]. Second-generation Muslims often face more challenges than their parents, who often maintain a strong ethnic identity [33]. Situations such as banning the covering of women’s faces is difficult for many in this group. From a European perspective, security is often the main concern, whereas from a Muslim perspective, such bans are seen as infringing on fundamental rights, leading to feelings of discrimination [31]. Negative portrayals of Muslims can further lead to more sympathy for violent radicalization [32].

A study examining 135 forensic biographies of terrorists in the United States showed that alongside changes in cognitive constructs, as described by the PCT, there are also shifts in behavioral patterns during radicalization processes [3]. This study identified four distinct phases of behavior within the radicalization processes. In general, radicalization processes begin with a pre-radicalization phase in which perception of deprivation, factors such as trauma, or other adverse circumstances prompt an individual to seek answers, belonging, or purpose in life [3]. Seeking general information about religion could provide answers to life questions [28]. In this phase, a cognitive opening occurs that increases vulnerability and susceptibility to new ways of thinking [3,34,35]. The pre-radicalization phase is often followed by the first phase, lifestyle adaptation, in which, for example, radicalized views are spread to inspire others [3]. The second phase, called extremist engagement, means that like-minded people are sought out to spend time with (both online and offline). Black-and-white and us-against-them thinking is mainly formed in this phase, and justification for violent acts is formed [3,28,34]. In the third phase, namely the preparation for violence phase, an individual will bypass inhibition mechanisms to engage in violent behavior [3,28,34]. These mechanisms are circumvented by ingroup and outgroup categorization and by distancing the ingroup from the perceived enemy [28]. Figure 2 provides a systematic overview of these four phases of the radicalization process. This model describes behavioral changes in radicalized individuals. The model does not provide indications about which individuals move to the next phase of behavior under which circumstances [3].

Taken together, certain critical factors, such as strain and uncertainty, play an important role and can open the door to violent extremist ideologies. When an alternative, radical belief system offers a sense of certainty or a perception of having nothing to lose because of limited social connections, it can potentially lead individuals towards engaging in violent extremist behaviors [10,18].

### 1.3. Differences in Radicalization Processes between Male and Female Violent Extremists

Men and women can both react to strains with anger. However, men have been shown to face other types of strains than women. Men are more often subject to financial strain or interpersonal conflict, whereas women more often have strains involving high levels of social control [36]. Anger in women is expressed more often with feelings of depression, anxiety, and guilt, whereas in men it is more often expressed with aggression [36]. Furthermore, females do have better prosocial skills due to greater verbal ability and differential socialization by peers and parents [37]. Previous studies have also identified other unique aspects of female violent extremists (e.g., [38,39,40,41]. example, young Muslim women in Western societies may experience feelings of being different [42] and often lack social bonds with family or friends [43], which can create a sense of insecurity. These feelings of disconnection from society can give rise to a desire for sisterhood [42,43,44]. When radical thoughts are shared, a different worldview can emerge and a social identity can be developed that creates us-against-them thinking [18,42]. This social learning [44] related to radicalization processes can make individuals more prone to violent extremism [45,46]. Group membership can thus influence how one thinks about individuals and the world and how one feels and behaves in that process [47]. Belonging to a group is also an important source of pride and self-esteem observed in left-wing [48] and right-wing women [49]. Next to this, women may also join violent extremist groups to gain status, pride, freedom, social identity, self-esteem, and a sense of purpose. Studies show that revenge after a significant loss is an important motivation for violent extremism in women [41,50], whereas ideological factors, improving social status, and monetary rewards are more common in men [39,41]. Especially in patriarchal societies, perceived gender inequality and feelings of strain are the driving factors for radicalization in women [51].

### 1.4. Current Study

Previous analyses of radicalization processes argue that a state of uncertainty can increase susceptibility to associating with radical groups, including us-against-them thinking, giving a new sense of certainty and recognition [18]. Radicalization processes include four phases of behavior patterns and can lead to justification of violent extremism against the other group [3]. The aim of this study is to investigate these underlying mechanisms of radicalization processes in female and male violent extremists. The main research question is: “Which transition points, involving changes in cognitive constructs according to the PCT and changes in behavioral patterns following the four-phase model, can be identified among individuals, both men and women, residing in Dutch terrorism wings?”. The second research question is: “What are the differences in these transition points between men and women?’’. As previously mentioned, prior research showed variances in the radicalization processes between men and women. It is hypothesized that women are more likely than men to experience social isolation and feelings of loneliness, whereas men are more inclined to pursue status and engage in ideologies legitimizing violence. Consequently, we thus expect that transition points for women will be related to feelings of loneliness and a lack of social connection, while those for men will be connected to a pursuit of status. It is noteworthy that a majority of existing research on violent extremism has focused on male samples [5]. Getting more insight into discrepancies in the radicalization processes between men and women can contribute to the development of gender-specific deradicalization interventions and policies to adequately respond to radicalization [52] and to prevent future violent extremism by women [7,8]. In this study, we adopted a person-centered perspective and assumed that trigger factors at the individual level can increase susceptibility to alternative worldviews and, in the case of radicalization and extreme ideologies, thereby initiate a process toward violent extremism [34]. To this end, we examined specific processes (changes in cognitive constructs and behavioral patterns) that may lead to violent extremism and addressed differences between female and male detainees residing in Dutch terrorism wings.

In contrast to traditional ‘variable-oriented’ (quantitative) approaches dominant in the social sciences [12,53,54], we opted for a person-centered (qualitative) perspective in which patterns of relevant personal characteristics could be investigated [21,54]. The person-centered perspective adopted in this study takes an exploratory and descriptive approach, focusing on examining the development of the radicalization process in individuals in a detailed manner rather than through quantitative methods [53]. In the current study, we have reconstructed the biographies of 13 male and 13 female detainees residing in Dutch terrorism wings. These reconstructions are based on retrospective file research, encompassing a wealth of data sources. Our dataset comprises criminal records, police reports, and probation reports. In addition, we had access to prison files, including mental health status and behavioral assessments, such as psychological and psychiatric assessments, staff observations, file information, Violent Extremism Risk Assessment reports, and information from relatives. Furthermore, the prison files included individual and group analyses conducted by a professional with extensive knowledge of cultural backgrounds, extremist ideologies, and the Arabic language. This expert actively participated in multidisciplinary meetings regarding the detainees and provided valuable insights into their behavior within the context of (sub)cultures, religion, and language. With these extensive sources of data, we were able to reconstruct the biographies of these radicalized individuals. This reconstruction made it possible to investigate person-environment interactions, sequences of actions or behavioral patterns, and individual change over time [53].

## 2. Materials and Methods

### 2.1. Study Sample

In the period 2014 to 2022, 14 female detainees resided in the Dutch terrorism wings of the Vught prison facility. Thirteen of them were suspected or convicted of jihadist violent extremist offenses. Ages ranged from 23 to 46 years (*M* = 31.62, *SD* = 6.31) at the time of placement in the terrorism wings. One right-wing extremist woman was excluded from the study because right-wing extremism was not the focus of the study, and its inclusion would harm the homogeneity of the study sample. To investigate gender-specific characteristics and transitions, 13 comparable jihadist men were selected from the total jihadist male population (*N* = 124) who stayed in the Vught facility between 2014 and 2022. These men were aged between 25 and 40 years (*M* = 31.15, *SD* = 4.08). The matching of female and male detainees was based on age in accordance with existing literature indicating differences in characteristics between foreign fighters and homegrown terrorists (e.g., [3]). They were also matched by the type of terrorist act for which they were suspected or convicted, because trajectories throughout the life course may vary depending on the type of offender [55,56].

The index crimes of the women and men together included twelve returned Dutch foreign fighters, eight individuals who attempted to travel to Syria or Iraq, two individuals involved in sedition or propaganda, two individuals involved in financing terrorism, and two individuals who were members of a terrorist organization. See Table 1 for a detailed overview of the socioeconomic characteristics of the sample.

### 2.2. Measures

*Radicalization process and transitions.* Central to the study was the reconstruction and analysis of narratives through a process of constant case comparison [57]. While applying this method, it is important that comparison is a frequent and constant process and not just a phase to be completed at the end of the analysis. See Section 2.4 data analyses for an in-depth description of the data collection form and the indicators that are used in this study.

### 2.3. Procedure

Violent extremist detainees have a special security status in the Dutch detention system. They can have a major societal impact because of their potential threats to public safety. Therefore, all violent extremist detainees undergo a standard comprehensive assessment. During this assessment, intra- and interpersonal dynamics of detainees are inventoried, and it is investigated what interventions are needed to reduce potential intramural safety risks and reduce the recidivism risk. The intrapersonal dynamics within a terrorism wing provide information on the group dynamics among detainees, such as identifying those susceptible to influence or recognizing charismatic leaders. The interpersonal dynamics pertain to shifts in an individual’s cognition and behavior changes over time or in response to different situations.

In this study, we used data collected as part of the regular detention process. This study was approved by the Ethics Review Board of Tilburg University. Because of the high confidentiality of the data, the Data Protection Office of Tilburg University checked the data protection and data management. Informed consent was not used because it was not possible to trace all detainees (at the time of study, most of them had been released from prison), and there was a high risk of selection bias because of the nature of the study content and population. Based on the General Data Protection Regulation and Dutch privacy legislation, the data can be used because this research serves a public interest. In this case, Article 14.5(b) applies because informing the detainees would interfere with the research and appropriate measures were taken to protect the detainees’ rights. The Ethics Review Board decided that a public statement was not required or desirable because of the sensitivity and relevance of this research and the potential danger of informing the subjects. All data were processed anonymously and are not traceable to individuals.

### 2.4. Data Analyses

Descriptive statistics were computed in IBM SPSS, version 28.0 for socio-demographic, psychopathological, criminal, and socio-cultural characteristics. Next, we reconstructed the biographies of 13 male and 13 female detainees staying at Dutch terrorism wings. From the available data (consisting of criminal and police records, probation reports, and prison files), we subtracted information according to an a priori list of subject characteristics (see Table 2). Our data extraction was guided by a predetermined list of subject characteristics (see Table 2). The indicators and categories in this format are drawn from previous research, specifically factors associated with the personal transitions and trajectories of radicalization and deradicalization [18], indicators from Violent Extremist Risk Assessment 2R [58,59], and findings from a systematic review on risk and protective factors of female violent extremists [60]. These studies were chosen for their exploration of relevant factors in radicalization processes, encompassing changes in cognitive constructs and accompanying shifts in behavioral patterns. Furthermore, these identified factors align with research on transition points in radicalization processes (e.g., [7,54]). The data collection format was discussed by the first and fourth authors and specified after discussion. See Table 3 for a specification per item.

This extraction process was performed by the first and fourth authors and involved a thorough cross-verification to ensure that no critical information was missing. To maintain consistency and employ transparent coding rules, each item was systematically described for every case, as outlined in Table 3. Furthermore, we adopted a constant case comparison approach, wherein we compared the items across all cases during the coding phase. The continuous comparison of cases involved comparing individual cases through the process of coding, analysis, and comparison of biographies, by identifying similarities and differences in biographies, and by detecting possible patterns. In this way, insights were gained into the themes under investigation, namely cognitive constructs and their transitions. Through this analysis, we compared the cases of the identified items and screened the factors contributing to the radicalization process (e.g., ‘’What were factors that contributed to the decision to travel to Syria?’’). As a result, we were able to position the identified factors, changes in cognitive constructs, and changes in behavioral patterns within the context of the timeline within the PCT model (see Figure 1; [18]). The construction of these timelines and their subsequent comparisons were overseen by the first and fourth authors. Additionally, we assessed the corresponding phase within the four-phase model (see Figure 2; [3]). Having established a comprehensive overview for each case, we compared female and male violent extremists. Next, we proceeded to compare the timelines between male and female violent extremists.

## 3. Results

### 3.1. Patterns in Radicalization Processes and Transition Phases in Female Detainees

Women reported that prior to their radicalization, they had little to no knowledge of the caliphate declared by ISIS. All but one were born in the 1980s/90s. About half of the sample (46.2%) were second-generation Muslim migrants. The sample could be divided into a group characterized by dysfunctional family dynamics (n = 8) and a group characterized by a stable family situation (n = 5).

Figure 3a provides an overview of characteristics of the radicalization processes and transitional phases of the first group. Dysfunctional family dynamics included coming from a broken home, large family responsibilities at a young age, lack of parental attention, or parents unable to care for their children. In several cases, there was domestic violence between the parents during childhood. This home situation had a negative impact on the feeling of being seen, heard, and valued. Women had a pre-radicalization phase in which there were feelings of uncertainty and a lack of social validation. In adolescence, this often manifested as identity problems with important questions such as: ‘Who am I?’ and ‘Where do I belong?’ For some, a lack of social validation resulted in isolation (sometimes related to the family, but often to society) and was reinforced by negative media coverage of Muslims through campaigns such as ‘the war on terror’. These situational factors strengthened ‘us against them’ thinking and created a negative construction of the ‘other’ group. This state of invalidity and uncertainty was increased when it coincided with stressful events such as a relationship break up, disturbed relationships with parents, and/or being expelled from school. Consequentially, they sought more secure worldviews, which created a more negative construction of the ‘other’ group and led to seeking out like-minded individuals. Women in this group were warmly welcomed into their (online) contact with radicalized peers or future partners. This initiated the phase of lifestyle adaptation (Phase 1). Further deepening and adherence to the Islamic rules of the faith had a soothing effect and offered structure, meaning, and belonging. However, this new way of life also created problems in relationships with friends and relatives who could not always appreciate or even disapproved of the new lifestyle. In some cases, women went through the second phase by legitimizing violence for the greater good (especially in the fight against the Assad regime). They then moved on to violent extremist actions (Phase 3), for example by joining a foreign insurgency or by providing nonviolent support for terrorism by fundraising or recruiting for terrorism. In some cases, this engagement in terrorism resulted in disappointment. Women who returned from Syria or Iraq often expressed disappointment, resulting in an invalidation of the Islamic caliphate and a lack of social validation by IS members. The utopian image of the caliphate was not as they had imagined it. For several women, situational factors were also responsible for their disappointment, such as marital tensions and dire living conditions (also for their children) and traumatic experiences during the war. Together with the cognitive construction processes, these situational factors led to disengagement from extremist violence. For some, re-establishing ties with old friends and family was a first step in disengaging from radicalization. Moreover, the desire for a different way of life, particularly focused on motherhood and education, may have helped in their disengagement process.

The group (n = 5) characterized by growing up in a stable family situation had different accents in their development than the other group (see Figure 3b for an overview). Their feelings of not being heard, seen, and valued were independent of the family of origin, but stemmed from an identity struggle, with questions such as: ‘Am I Dutch or something else?’ The families of two of the women in this group belonged to a distinct minority group in their country of origin and were oppressed there. These families came to the Netherlands as political refugees. The feeling of being a social outsider had been an issue within these families for generations. This created feelings of uncertainty and formed a breeding ground for radicalization, the so-called pre-radicalization phase. This group became disappointed in the Dutch government because, in their opinion, the government did nothing about discrimination against Muslims. This created a negative image of ‘others’, which was reinforced by negative media coverage of Muslims. Seeking recognition and support, all five women came into (online) contact with radicalized peers. Moreover, these women never married the radicalized peers with whom they interacted, as was the case for several women in the other group. There was also clear indoctrination by radicalized peers. So, the foregoing reflects the first radicalization phase, and the lifestyle adaptation was clearly visible. However, the justification for the use of violence was not reflected in the radicalization processes of this group. Hence, engagement in violent extremism was lower compared to the other group. This group seemed to seek more emotional support from radicalized peers and was indoctrinated more often than the other group. Some prepared for extremist actions, such as joining a foreign insurgency. The disengagement process in this group was influenced by the disillusionment of living in the Caliphate and the desire for a different way of life, focusing on motherhood, education, and re-establishing contact with family members.

### 3.2. Differences in Radicalization Processes and Transitional Phases between Females and Males

The radicalization processes in the female group differed from those in the male group in a few aspects. The latter were often known to the police and justice system prior to the terrorism-related offense (53.8%), whereas most of the women were first offenders (84.6%), which was certainly the case with younger detainees. Six males grew up with dysfunctional family dynamics (see Figure 4a for an overview) and seven grew up in stable home situations (see Figure 4b for an overview). Low self-esteem, perceived discrimination, and social isolation were relatively common themes in their radicalization process. Feelings that the Dutch government allowed discrimination often led to hostility towards the government. Feelings of uncertainty and invalidation were also important factors related to radicalization. Making efforts to right old mistakes (including previous police contacts) seemed to be an important factor among men to make lifestyle changes in Phase 1, which was not seen in women. Men in this group often saw religion as guiding their (new) life, which provided a more certain worldview. Compared to the female group, the desire for appreciation or status was more characteristic of men. Instead, women reported a stronger desire for reward and emotional support. Women were driven more by experienced trauma, feelings of loneliness, sibling indoctrination, and unsatisfying love relationships. When comparing factors contributing to disengagement, women had a greater need for self-sufficiency and education than men.

## 4. Discussion

This study used the life-history approach to uncover turning points and processes of change within the radicalization processes of female and male detainees residing in Dutch terrorism wings. Using a ‘person-oriented’ approach [21,54], we investigated whether certain patterns are visible in the transition points within the radicalization processes of these detainees and whether there are differences in these transitions between men and women.

Regarding the patterns in the transition points within the radicalization process in women, we found that person–environment interactions are relevant in understanding violent extremism.

Personal Construct Theory describes individual and social processes of change during the life course [18] and is helpful in understanding radicalization processes and transitional phases. Most women were adolescents when the 9/11 attacks occurred and the war on terror (including the invasion of Iraq, the Arabic Spring, and the Syrian conflict) became something they grew up with. After these attacks, Muslims became the target of negative prejudice (e.g., negative media attention around Muslims) and were confronted with discrimination [61], which created an uncertain worldview [18] and a breeding ground in the pre-radicalization phase [3]. We found that there were two subgroups in the female group. Women who came from a dysfunctional family system showed more experienced trauma in the pre-radicalization phase compared to women who came from a stable family. Previous research suggested that stressful family situations may indirectly influence the radicalization process [62]. When children or adolescents do not feel welcome or safe in their families, they may reach out to other groups that do provide this safety via affection, emotional support, attention, and recognition. For women who came from a stable family, the state of uncertainty did not have roots in the family of origin, but women in this group also had negative images of the ‘other’ group (e.g., Dutch government) because of experienced injustice against Muslims.

In line with previous research, we found that in the lifestyle adaptation phase (Phase 1), women find it difficult to bond with the society in which they live, creating a feeling of social isolation [18]. Women are thus looking for a community where they are accepted and where members stand up for each other. In some cases, radicalization can be seen as a coping mechanism to deal with problems or conflicts in their daily lives. A radical, more structured, and specific worldview seems to reduce complexity and can accommodate inner conflicts, while also ensuring belonging to a group of like-minded peers [18,63]. In this phase, we found that women from a stable family were more often indoctrinated by radicalized peers than women from dysfunctional family systems. A consideration is that it was easy to feed feelings of uncertainty because a negative image of the Dutch government was already present. Another consideration is that it took more indoctrination to increase feelings of insecurity to create enough of a breeding ground for the radical ideas of the radicalized peer group.

In the extremist engagement phase (Phase 2), we found that being part of a radical network (of friends of family members) was a very important factor for most women. This is in line with the assumption that radicalization is a social phenomenon and that it happens with like-minded people who support and influence each other [18,64]. Women from dysfunctional family systems showed more frequent justification for using violence to obtain goals compared to women from stable families. Spending time with like-minded people could almost be considered a prerequisite for motivating actions (Phase 3; [3]).

Interestingly, in several cases we found that the factors that led to radicalization later turned out to be factors that led to disengagement [21]. What previously provided support or hope (e.g., promotion videos of the Caliphate) could later lead to disappointment over unfulfilled expectations. Furthermore, becoming a mother was an important factor for disengagement in several cases.

Regarding the second research question about differences in transition points between women and men, the results show that women have a stronger need to belong, whereas men are more seeking recognition or appreciation. The men in this study often suffer from a negative self-image that they hope to improve by joining the caliphate as a soldier. We found that men were often known to the justice system prior to the terrorism offense. Previous research suggested that earlier contact with police is related to a higher risk of developing sympathies for violent extremism [65]. It is likely that the previous police contact in those cases contributed to the formation of cognitive schemas and increased feelings of uncertainty. In contrast, women in our group were more influenced by experienced trauma and feelings of loneliness. Taken together, differences between men and women were small. For women and men, belonging to a group was an important pull factor and both found it difficult to bond with the society in which they live. Feelings of loneliness and being different in Western societies can lead to insecurity and a need for social inclusion and attraction to violent extremist groups [42]. According to PCT, several factors (e.g., perceived grievances) contribute to feelings of insecurity [18]. When these factors become more prevalent, feelings of insecurity increase and can lead to adherence to more radical beliefs. These radical beliefs (and negative cognitive schemas belonging to these beliefs) can be further validated through contact with like-minded others and can lead to violent extremist behavior [18]. These underlying mechanisms are the same in female and male detainees residing in Dutch terrorism wings.

## 5. Limitations

Although this research provides clear indications of initial radicalization processes in both women and men, it is not free of methodological limitations. First, retrospective research was used and therefore the browsed content determines the quality of the study. In some cases, little information was available because the detainee was only in detention for a short time or because the detainee came from abroad and thus there was less prior information or fewer police files available. Another limitation is the small group, which makes it impossible to generate causal claims and generalize findings to the overall population of suspects or individuals convicted of terrorism-related offenses. Furthermore, this study included suspects of terrorism-related offenses who had not been convicted, which means they are in different stages of their criminal process that may affect the results. However, it is known that a substantial proportion of suspects will eventually be convicted. Furthermore, both suspects and convicts went through processes of radicalization.

### Implications for Policy and Future Research

The results of the current study showed that the construction of cognitive schemas along with (life) events is important in understanding radicalization processes. Prevention strategies should target individuals with uncertain worldviews with unfavorable situational factors that may cause turning points to radicalization. In the pre-radicalization phase lies the greatest potential with regard to the prevention of further radicalization which could result in violent extremist behavior. In women, and formulated with caution, experienced trauma seems to play a greater role than in men, while men seem to be more influenced by righting old mistakes and status concerns. Interventions targeting women should therefore focus more on emotional recognition of belonging and on experienced trauma, whereas interventions targeting men should focus more on moral reasoning and improving self-esteem so that they become less dependent on the status given by others.

Future research should look at life events from multiple perspectives to understand and stop the development of radicalization (and subsequent violent extremism). In addition, more attention to individual resilience could be an important avenue of further research because this could give more insight into helpful prevention strategies. In addition, investigating individual differences with a particular focus on adolescence and emerging adulthood as critical life stages could be interesting for future research. Finally, attention should be given to ‘religion as coping’. We showed that many individuals referred to religion as guidance, or a handhold for life, and that it brought them serenity and structure. Future research is needed to investigate whether specific disengagement interventions could be used to address this coping strategy.

## 6. Conclusions

While research on terrorism-related behaviors is emerging, empirical research is limited. From a qualitative perspective, radicalization processes in females and males have been studied and compared in detail in a group of suspects and convicts of terrorist crimes residing in Dutch penitentiary terrorism wings. We found two groups of women, namely a group that grew up in stressful family environments and lacking emotional support from their family and a group that did not experience such conditions. We also found differences in radicalization processes and transitional phases between women and men, which can be helpful in developing gender-specific prevention and disengagement interventions.

## Figures and Tables

**Figure 1 behavsci-13-00877-f001:**
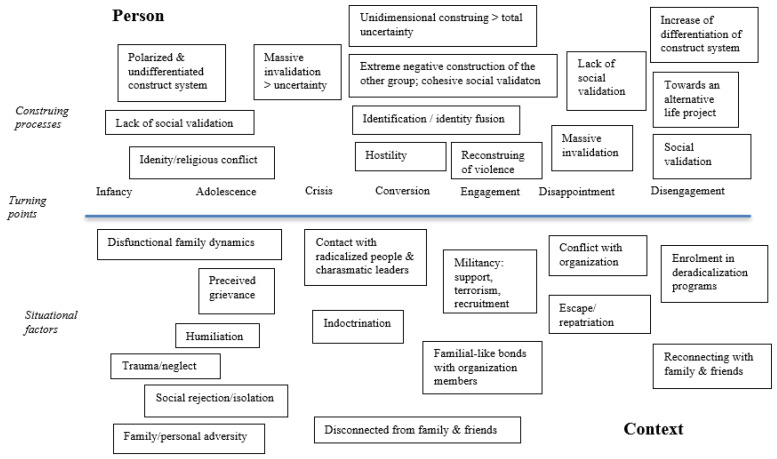
Personal construct heuristic model for process of radicalization and deradicalization [18].

**Figure 2 behavsci-13-00877-f002:**
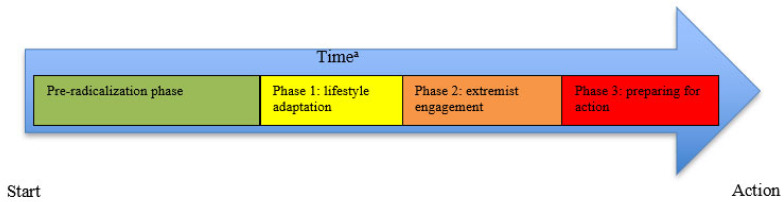
NYPD-model [3]. ^a^ Duration of phases differs per individual.

**Figure 3 behavsci-13-00877-f003:**
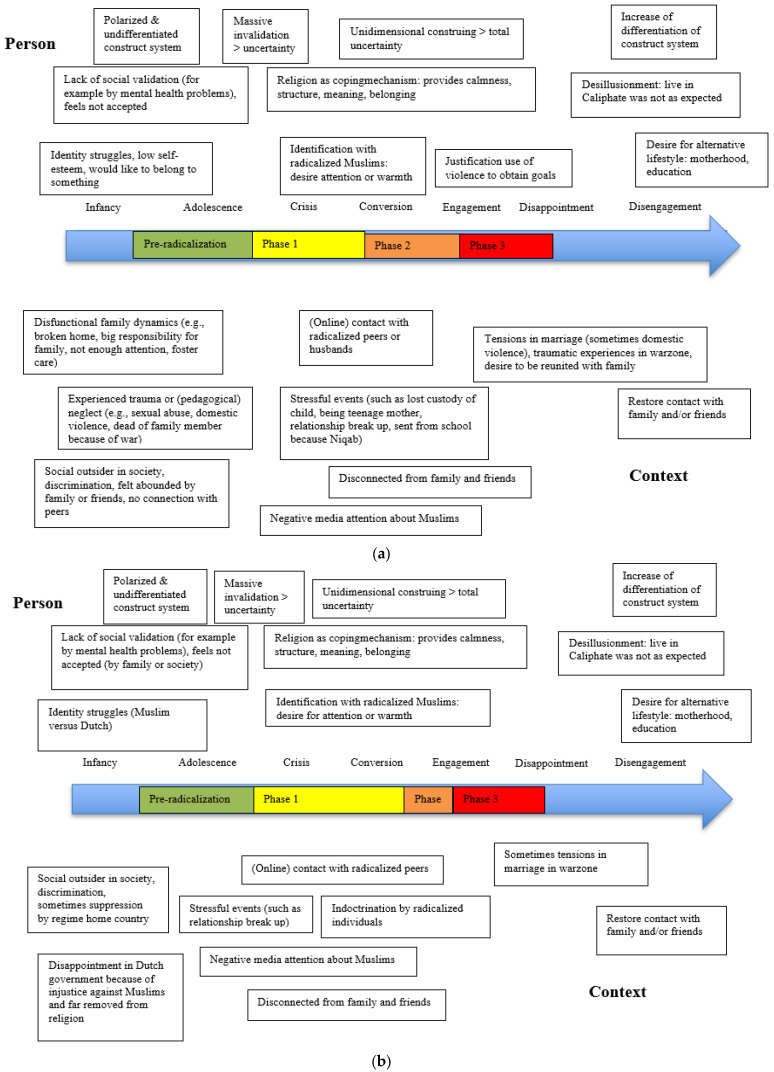
(**a**) From dysfunctional family dynamics to radicalization (n = 8) in women. (**b**) From a stable home to radicalization (n = 5) in women.

**Figure 4 behavsci-13-00877-f004:**
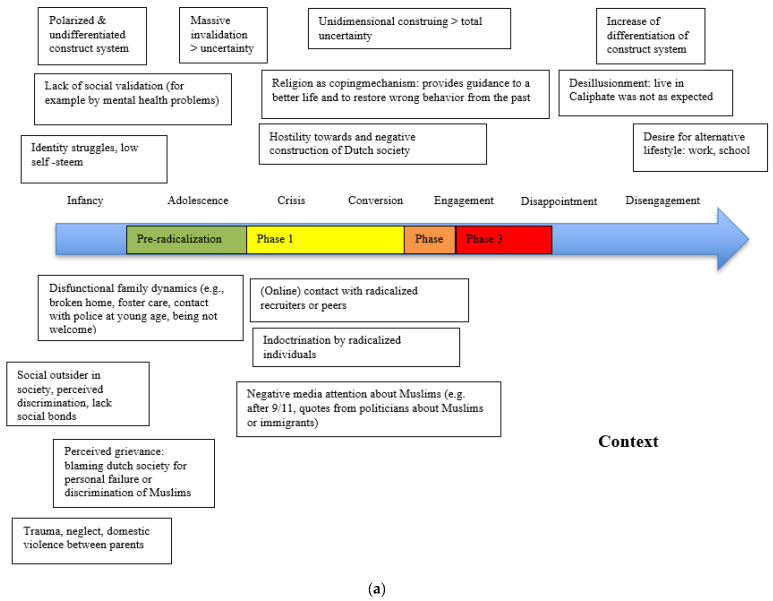
(**a**) From dysfunctional family dynamics to radicalization (n = 6) in men. (**b**) From a stable home to radicalization (n = 7) in men.

**Table 1 behavsci-13-00877-t001:** Characteristics of women compared to men on generic demographic information.

Variable	Male (*n* = 13)	Female (*n* = 13)
Average age	31.2 years	32.4 years
Born in		
Netherlands	61.5%	69.2%
Morocco	15.4%	0.0%
Middle East	23.1%	15.4%
Russia	0.0%	7.7%
Romania	0.0%	7.7%
Divorced parents		
No	61.5%	61.5%
Yes	38.5%	38.5%
Criminal antecedents		
No	46.2%	84.6%
Yes	53.8%	15.4%
Violent crimes	30.8%	7.7%
DSM-5 diagnosis		
No	62.0%	77.0%
Yes	38.0%	23.0%

**Table 2 behavsci-13-00877-t002:** Data collection form.

Item	Source	Response
General variables		
Gender	TULP	
Date of birth	TULP	
Index delict	TULP	
All available sources, mention all sources		
Personal variables		
Education	JD online	
Religion/ethnicity	TULP	
(Number of) siblings	JD online	
Marital status	JD online	
(Number of) children	JD online, justice file, prison file	
Generation migrants	JD online, justice file	
Behavioral/mental health problems	Justice and prison file	
Criminal antecedents	JD online, justice file	
Housing situation time of index delict	JD online, justice file	
Devotion to ideology that justifies violence	JD online, justice and prison file	
Religious conflict and seeking (new) religion/radical beliefs	Justice and prison file	
Negative feelings (insecurity, uncertainty, loneliness, helplessness)	Justice and prison file	
Heroism	Justice and prison file	
Hostility (national identity)	VERA-2R/Justice file	
Rejection social norms/values	VERA-2R	
Adoption other norms/values	JD online, justice and prison file	
Search for significance/meaning (making a difference)	VERA-2R	
Search for status (pride, self-importance, recognition)	VERA-2R	
Search for sensation/excitement	VERA-2R	
Search for revenge	Justice and prison file	
Search for identity/identity struggles	Justice and prison file	
Expressed emotions (frustration, anger) in response to perceived injustice	VERA-2R	
Seeking rewards (emotional, moral, monetary)	JD online, justice and prison file	
Crisis (personal, socio-cultural, economic)	JD online, justice and prison file	
Loss of significance (e.g., loss of loved ones)	JD online, justice and prison file	
Humiliation	JD online, justice and prison file	
Lack of empathy and understanding for the other group/negative construction of the other group	VERA-2R	
Susceptibility to influence, direction or indoctrination	VERA-2R	
Positive prospects (e.g., mothers-to-be, employment, stability)	JD online, justice and prison file	
Desire normative lifestyle	JD online, justice and prison file	
Sufficient status/purpose	JD online, justice and prison file	
Situational variables		
Dysfunctional family dynamics (broken or loose family, lack of affection from parents, divorced, conflicted, or absent parents)	JD online, justice and prison file	
Network of family/friends/acquaintances involved with (extremist) organization	VERA-2R	
Personal contact with violent extremists	VERA-2R	
Anomie/strain (e.g., patriarchy, economic situation)	JD online, justice and prison file	
Experienced trauma/neglect	JD online, justice and prison file	
Social rejection/isolation/unacceptance	JD online, justice and prison file	
Familial or personal adversity (e.g., discrimination, relationship problems)	JD online, justice and prison file	
Disconnection from family and friends	JD online, justice and prison file	
Family with traditional (gender) roles	JD online, justice and prison file	
Escape/repatriation	JD online, justice and prison file	
Search for camaraderie, belonging to a group, social needs/idea of a Utopian society	VERA-2R	
External threat to (Muslim) community	JD online, justice and prison file	
Perceived grievances/injustice	VERA-2R	
Political crisis/social disorganization	JD online, justice and prison file	
Cognitive opening (due to community crisis)	JD online, justice and prison file	
Imminent existential threat (community)	JD online, justice and prison file	
Coercion	JD online, justice and prison file	
Gender (in)equality	JD online, justice and prison file	
Reconnecting with family and friends	JD online, justice and prison file	
Prosocial surrounding/inclusion/new partner or friend	JD online, justice and prison file	
Support from family members or significant others for nonviolence	VERA-2R	
Community support for Nonviolence	VERA-2R	
Herinterpretation ideology	VERA-2R	
Rejection of violence to achieve goals	VERA-2R	
Herinterpretation of the enemy	VERA-2R	
Participation deradicalization programs or groups	VERA-2R	

**Table 3 behavsci-13-00877-t003:** Specifications of items in data extraction format.

Item	Specification
General variables	
Gender	Male/female/other
Date of birth	Date of birth
Index delict	(Terrorist) offense for which the individual is placed in prison
All available sources, mention all sources	
Personal variables	
Education	Specify highest level of education. Degree obtained or not obtained. Specify difficulties in school or with learning.
Religion/ethnicity	Cultural background of the individual/their family. Religious background of the individual/their family.
(Number of) siblings	Total number of siblings, describe also half brothers and sisters.
Marital status	Specify: married, divorced, engaged, single, relation.
(Number of) children	Total number of children.
Generation migrants	Specify generation of migrant (first, second, third). Asylum seeker. Status holder.
Behavioral/mental health problems	Describe behavioral or mental health problems. Previous diagnoses and treatments. Onset, duration, coping mechanisms, social support regarding these complaints. Describe situational circumstances in which the complaints take place.
Criminal antecedents	Describe previous criminal antecedents. Previous imprisonments or other sanctions. Also describe the context in which the antecedents take place (e.g., with others, during drugs abuse).
Housing situation time of index delict	What was the housing situation during the terrorist offense?
Devotion to ideology that justifies violence	Describe whether the individual adheres to an ideology that is well justified. Does the individual see any limitations to the use of violence to achieve ideological goals? How long has he/she had these beliefs? How obliged does someone feel to violence?
Religious conflict and seeking (new) religion/radical beliefs	Does someone experience a conflict with their current religion (e.g., non-practicing Islamic upbringing but wanting to practice). Does anyone experience a difference between what society expects versus what faith expects?
Negative feelings (insecurity, uncertainty, loneliness, helplessness)	Does the individual experience negative feelings at any time (such as fear, uncertainty, loneliness, helplessness)? Describe these moments and the feelings. How does the person deal with these feelings? Are there changes in how the person subsequently views themselves, others, or the world? Does the person involved experience support with these feelings? How does the person involved interpret these feelings (e.g., are they due to a certain situation or person)?
Heroism	Does the individual feel that he/she is above others because, for example, he/she has more knowledge about faith or is a believer (while others are non-believers)? Does he/she feel superior, and does he/she believe that certain behavior is therefore justified?
Hostility (national identity)	Does the individual identify with the national identity of the country where he/she resides? Does he/she experience any dissatisfaction/distance about national identity? Is the individual hostile to national identity?
Rejection social norms/values	Does the individual accept the social norms and values of the democratic and pluralistic society in which he/she lives? Does he/she feel part of this society even though he/she may be against its norms and values? Does the individual reject certain democratic values and norms? Does an individual distance him/herself from certain democratic values and norms?
Adoption other norms/values	To what extent is an individual involved in other norms and values (such as those of a radical group/Sharia law). To what extent are other values and norms adhered to (and are they in conflict with democratic values and norms)? To what extent does this cause conflicts with, for example, loved ones or institutions (such as school or work)?
Search for significance/meaning (making a difference)	Is an individual motivated to participate in (violent) extremism or to be part of such a group with the feeling of making a difference or being of significance? Were there feelings of emptiness or uselessness? Does anyone feel like contributing to a higher purpose?
Search for status (pride, self-importance, recognition)	Is an individual motivated to participate in (violent) extremist acts out of a need for status? Did he/she previously gain prestige through (violent) actions? Did someone previously feel worthless and wanted to eliminate that feeling by joining a violent extremist group?
Search for sensation/excitement	Is an individual motivated by excitement and adventure to participate in extremist (violent) acts? To what extent does someone like excitement in his/her life?
Search for revenge	Was there a significant loss? Was this loss attributed to a specific person or group of people? This may include deceased loved ones, but also the infliction of damage by, for example, a government on Muslims. Does an individual participate in violent extremism to correct harm?
Search for identity/identity struggles	Is an individual motivated to participate because he/she is searching for who he/she is or where he/she belongs (e.g., Muslim versus European)? What are his/her norms and values? Does a radical ideology provide answers to these life questions?
Expressed emotions (frustration, anger) in response to perceived injustice	Does the individual express anger, moral outrage and/or despair as a result of feelings of perceived injustice (both individually and from a group context)? What feelings are there towards those who threaten their beliefs?
Seeking rewards (emotional, moral, monetary)	Is there a search for reward? This reward can be emotional (clearly looking for emotional support), moral (looking for like-minded people) or monetary. Describe what is happening and also describe the context in which this takes place.
Crisis (personal, socio-cultural, economic)	Is an individual feeling a high level of stress/uncertainty (crisis)? This can be personal (for example, because negative life events come together), socio-cultural, or economic. Describe the crisis as experienced by the individual. Also describe how someone dealt with it and whether there were adequate coping mechanisms.
Loss of significance (e.g., loss of loved ones)	Is there a clear experience of loss of something that was really important to that person (e.g., the death of a loved one, loss of a valuable job).
Humiliation	This can be personal or at group level (e.g., against all Muslims)? This involves a feeling of humiliation. Describe this feeling and also describe whether and, if so, who is held responsible for this humiliation experienced.
Lack of empathy and understanding for the other group/negative construction of the other group	Does the individual lack empathy and understanding for people outside their own (cultural, religious, or ideological) group? Is there rigid us/them thinking? Is there moral detachment from people outside one’s own group?
Susceptibility to influence, direction, or indoctrination	Is an individual susceptible to influence or direction by a leader or person who promotes the use of extremist violence? Has the individual often been very influenced by others?
Positive prospects (e.g., mothers-to-be, employment, stability)	Is there a positive outlook for the future, such as for example having a child, work, home, re-establishing contacts with old friends or relatives.
Desire normative lifestyle	Wanting to live a ‘normal’ life can be a protective factor against violent extremism. Does the individual have the desire to live a ‘normal’ life? Does he/she want to focus on the role of parent to their children? Does he/she want to finish school or find a job?
Sufficient status/purpose	Is there a sufficient sense of status and purpose in life? Does an individual feel that he/she is useful or valuable? Is the self-image sufficiently strong?
Situational variables	
Dysfunctional family dynamics (broken or loose family, lack of affection from parents, divorced, conflicted or absent parents)	Does the individual come from a family with dysfunctional family dynamics? Examples include divorced parents, emotional neglect, arguments between parents, absent parents, alcohol, or drug use by parents).
Network of family/friends/acquaintances involved with (extremist) organization	Does the individual have a network of family, friends, or associates involved in violent extremist organizations? Who is it about? What is the nature of the relationship? What does it mean for the individual that a loved one is involved in such an organization?
Personal contact with violent extremists	Does the person concerned have contact with violent extremists? Is this online or offline? How many people are involved? What is the relationship to these people? How sustainable are the relationships? Where did he/she meet this person?
Anomie/strain (e.g., patriarchy, economic situation)	Is there strain due to, for example, the patriarchy of the economic situation of a country?
Experienced trauma/neglect	Does the individual have traumatic experiences throughout his or her life? Is there emotional or physical neglect? Describe the experiences. How did the individual deal with it? What meaning has the individual given these events?
Social rejection/isolation/unacceptance	Is there social rejection, isolation, or unacceptance during life? Is the individual part of a minority group? Has there been bullying behavior? Or a strong feeling of loneliness?
Familial or personal adversity (e.g., discrimination, relationship problems)	Did the individual experience personal setbacks (such as relationship problems, losing a job)? Were there any setbacks within the family? Also consider a feeling of being disadvantaged/discriminated by others.
Disconnection from family and friends	Is there a disconnect or problems in the relationship with family and friends? For example, by gaining other (radical) friends or by no longer agreeing on norms and values, which causes arguments. Describe the nature of the disconnection/problems.
Family with traditional (gender) roles	Are there clear and traditional gender roles within the individual’s family and family? Describe what these looks like and what their effect is on the individual. What does the individual think of these roles? Does he/she experience limitations because of this?
Escape/repatriation	Is there an escape or repatriation from a violent extremist group or from Syria? Is the person involved happy with this? How long has he/she wanted this? Why? Has someone been forcibly placed in a re-education camp? How does he/she look back on this?
Search for camaraderie, belonging to a group, social needs/idea of a utopian society	Is the individual motivated by a desire for camaraderie and belonging to a group to participate in extremist acts? Is the person concerned sensitive to social pressure? How important is it to be part of such a group and what benefits does it bring to the individual?
External threat to (Muslim) community	Does the individual or the group to which he/she belongs experience a strong external threat against their in-group (Muslims)? Who is causing the threat?
Perceived grievances/injustice	Does the individual have grievances or perceived injustice about political, religious, or social matters (either individually or from a group context)? What do these grievances consist of? Who causes these?
Political crisis/social disorganization	Is there a political crisis or social disorganization? Consider countries where civil wars take place.
Cognitive opening (due to community crisis)	Is there a cognitive opening for violent extremism because there is a community crisis? Describe the cognitive opening and the context.
Imminent existential threat (community)	Is there an imminent external threat in the community the individual lives in? Describe the imminent external threat and the situation in which it occurs.
Coercion	Is there forced participation in extremist acts of violence? There may also be some degree of coercion. Consideration may also be given to a significant threat from, for example, a partner or a leader if a refusal to perform certain behavior occurs.
Gender (in)equality	Does the individual have clear gender (in)equity? Describe how any gender differences in the environment of the individual are viewed and what the individual’s view on this is. Is he/she satisfied with this or not?
Reconnecting with family and friends	Has contact been restored with family and friends? With whom? What does this look like? Who took the initiative? How important are these people to the individual?
Prosocial surrounding/inclusion/new partner or friend	Is there a (renewed) prosocial surrounding? Does the individual feel like he or she belongs somewhere? Is there a new partner or friends?
Support from family members or significant others for nonviolence	Does the individual experience support from family members or significant others for nonviolence or for leaving a violent extremist group? What does this support look like? Who gives it?
Community support for nonviolence	Is there support from the community in which the individual finds himself for non-violence or for leaving a violent extremist group? What does this support look like? Who gives it?
Herinterpretation ideology	Is there a change in values about extremist and rigid ideology; is the individual considering a new interpretation of his/her ideology? What does this look like then? What makes someone consider this reinterpretation?
Rejection of violence to achieve goals	Does the individual use non-violent means to realize his/her vision and does he/she reject that violence is justifiable to achieve ideological goals? To what extent does someone do this?
Herinterpretation of the enemy	Is there a change in the individual’s image of the enemy? Is the individual open to considering new alternatives regarding the enemy image? How did this change come about?
Participation deradicalization programs or groups	Has the individual (voluntarily) participated in deradicalization interventions? If so, was this voluntary? What were they made of? How does he/she look back on this? Has this caused a significant change in constitutions or behaviors?

## Data Availability

Information about the data set, data selection, and the data protection process and checks can be obtained from the first (and corresponding) author.

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
