# Peer review of "Radicalization Processes and Transitional Phases in Female and Male Detainees Residing in Dutch Terrorism Wings"

_behavsci, 2023, doi:10.3390/bs13100877_

Round 1
Reviewer 1 Report
This article uses case-level data to provide a qualitative assessment of whether, and in what ways, the process of terrorist radicalization differs in females versus males. Key results suggest that trauma and a lack of social support may be more common and salient for female radicalization, and that prior criminal record and status-frustration may be more common and salient for male radicalization. I think the study yields interesting and potentially important results, but I likewise think that the manuscript would benefit greatly from (a) a clearer articulation of precisely how and why the literature cited in the front end frames the overall goals of the analysis, (b) greater specification of important methodological details that remain somewhat unclear in the methods and results sections, and (c) the inclusion of potentially important literature that is not currently referenced.
With respect to the first of the above issues, I think the core features of the PCT model could more clearly and more specifically be provided for the reader. Section 1.1 describes the model as one in which individuals “give meaning of their world in response to life events through cognitive schemas created to make sense of the world and give meaning to our experiences.” This statement, however, seems both tautological and vague. If my interpretation is correct, the PCT model presumably is one in which life experiences are said to affect individual’s sense of meaning. This assertion seems reasonable and likewise seems relevant to radicalization and deradicalization, but it remain unclear (to me) from the remainder of this section: (a) whether adversity, rejection, and grievance are -the- primary factors that the model suggests to affect negative emotions or whether these are intended only to be examples of the types of factors that may do so; (b) the precise conditions under which the model predicts that adversity, rejection, and grievance will result in radicalization; (c) what the authors mean by suggesting that “By holding on to more radical (ideological) beliefs and prejudices, ‘turning points’ can be integrated into one’s identity with a more structured and secure view of the world”; and (d) what set of circumstances the model suggests to be indicative that violent actions have improved an individual’s sense of self enough to affect the likelihood of acting on radical beliefs.
On the one hand, section 1.2 arguably goes on to address some of the questions I reference above concerning section 1.1 but, on the other hand, much of the information provided in section 1.2 remains underspecified and unclear to me. The model presumably suggests that the development of a new identity and/or radicalization are “the result of multiple choices and transformations that can take place within a certain period of time,” but what specific choices are salient in the model, and what specific period of time does the model suggest to be required and/or likely? This section further suggests that radicalization takes place “step by step,” but do the authors simply mean to suggest that it takes place in the four stages/phases described, or do they intend “step by step” to mean something else? Further, this section suggests that the pre-radicalization stage “is often followed” by the lifestyle adaptation stage, but under what conditions (if any) does the model suggest that pre-radicalization will or will not be followed by the lifestyle adaptation stage? If the answer is simply “factors like deprivation, trauma, and other adverse circumstances,” does the model provide any specific guidance as to how much of these things will (or will not) tend to lead an individual to move on to the next phase? Finally, section 1.3, which is presumably intended to delineate differences between male and female radicalization described in prior research, technically speaks relatively little to these differences, with essentially only the final two sentences actually addressing male/female differences. My reading of the existing literature is that there is much more written about male/female differences than what these two sentences and these limited citations describe. In sum, then, I think that providing more precise details about the predictions of the PCT model would be useful for helping the reader determine how much the data presented later do or do not support the accuracy of the PCT model and, simultaneously, I think that section 1.1 could usefully begin with a clearer articulation of the model’s core essence before then moving on to incorporate further detail about the model’s predictions (beyond what detail is currently provided).
With respect to my second big-picture suggestion, it seems to me that much more detail could usefully be incorporated into the methods section about exactly what data were available in the case files described, as well as about the precise methodological procedures the authors used to analyze the data. For example, the author states that their data came from “narratives based on retrospective file research,” but it remains unclear to me whether different people may have collected data related to different cases. If so, might different people have systematically provided different types of narratives and might there be any way to evaluate the consistency/reliability of the procedures that different people used in assembling case files? Likewise, what exactly is it about a “person-centered” approach versus a “variable-centered” approach that the authors believe make it possible to examine person-environment interactions with the former but not the latter? More critically still, what did the authors’ “retrospective file research” actually involve, what did a typical case file include within it, what does the process of “constant case comparison” involve, and what specific coding decisions did the authors use to determine what types of raw data from the case files translated into what specific parts of the “data format” that the authors constructed to guide their analyses. I recognize that the information provided in Table 2 is likely relevant to the last question, but I nonetheless reiterate that I do not recall a dedicated discussion in the methods section about what coding protocol and/or what coding rules the authors used to determine whether something in the narrative files should or should not count as, say, “religious conflict,” “rejection of social norms,” etc. Likewise, I recognize that some of my above questions are intended to be answered by section 2.3, but even this description seems vague and somewhat unclear. What, for example, do the authors mean by “individual and group analyses by an expert on cultural aspects” and who was the ostensible expert in question? Similarly, in section 2.4, what does it mean to say that authors “placed the visible interactions in the four-phase model?” As I mention above, I remain uncertain as to what coding rules the authors used to determine what pieces of data should be coded in what specific ways. Absent a clearer sense of these coding decisions, it remains difficult to evaluate whether the results described in section 3 (interesting as I find them) are reasonable in light of the available data.
Finally, apart from my previously-mentioned suggestion that more literature about male/female differences in radicalization could usefully be incorporated into the literature review, it strikes me that a large body of potentially-relevant literature concerning criminological theory is potentially misrepresented in the present manuscript, or omitted altogether. Agnew’s 1992 article in Criminology, for example, wasn’t really about patriarchy or gender inequality, let alone about radicalization among females, and Agnew’s 1997 article with Lisa Broidy, which was at least about male/female differences in crime from a GST perspective was not referenced at all; nor, for that matter, was Agnew’s explicit treatment of terrorism (rather than crime/delinquency) from Theoretical Criminology (2010?). The latter citation seems particularly important because it struck me throughout the manuscript that Agnew’s GST framework for the study of terrorism seems at least as relevant to the present purpose as the model that the authors have chosen. Similarly, my recollection of Moffitt’s Developmental Taxonomy was that it did not concern different types of offending, per se, so much as it concerned different types of offenders, whose trajectories across the life-course are different from one another’s. Likewise, many of the arguments that the authors make about adult social bonds and turning points seem precisely in line with Sampson and Laub’s life-course theory of social control (1993) and with extensions/modifications that Sampson and Laub have since made to their original treatment of this theory. Finally, given that a substantial percentage of the study sample appear to have been second-generation immigrants, I think it would be useful for the authors to consult the existing literature about why second-generation immigrants are particularly prone to radicalization. In short, a number of scholars have observed that they are not as tied to their parents’ country of origin as are their parents (and often weren’t even born in their parents’ country of origin at all), but they similarly feel alienated in their current country as a result of discrimination toward immigrants (even as those immigrants perform labor that local nationals do not want to perform). Given that they did not have any choice in their parents’ decision to immigrate, this puts second-generation immigrants (i.e., the children of first-generation immigrants) into a difficult position and, potentially, makes them particularly amenable to recruitment. I think Jason Burke may talk about this to some degree in The New Threat, but I have seen this important argument discussed by other scholars as well. The authors might wish to consider whether the immigration status of many in their sample might make it worthwhile to mention these issues somewhere in the manuscript.
I think the quality of the English was generally good, although I think that some of my comments/suggestions might be usefully addressed (in part) with some further editing to the language for clarification.
Author Response
Dear reviewer,
Please find attached our revised manuscript for Behavioral sciences. We thank you for the helpful comments on our manuscript. The changes to the manuscript are described in more detail below. In addition, we have highlighted the changes to our manuscript within the document using colored text. We hope that we have revised the manuscript to your satisfaction and look forward to your reply.
Kind regards,
On behalf of all co-authors,
Gaby Thijssen

Reviewer 2 Report
The objectives of the study must be more clearly elaborated. The authors said that the question they want to answer is whether there are any patterns in the transition points within radicalization processes. But in what sense? Patterns of what? Why should we expect them, and what should they show us? What should they reflect anyway? That remained completely unclear. The second, related question is therefore equally unclear: are there differences in these transition patterns? In what sense? What differences are possible here? What should we expect? Without these specifications, the further elaboration of the results seems ad hoc and insufficiently structured.
Similarly, some clarifications are necessary regarding the research methods and design. First of all, the meaning of the categories listed in Table 2 is impossible to understand without at least minimal clarification of their content. What does, say, the category "house situation time of index delict" mean? And "Heorism"? Or "gender (in)equality"? It is impossible to understand what these categories refer to and how the material is coded. This must be described in more detail, for example, as an appendix.
In addition, source information is missing for a good part of the categories. How should it be understood? Does this mean that those categories were not even used? Then why are they listed at all? Furthermore, it is unclear how this category list or coding scheme was created in the first place. Why these and this number of categories? Was their creation guided by earlier research? Theoretical expectations? It was not clarified. Finally, it is not clear how the categories are populated. What was the coding process like? Who did it and how? It was only mentioned in passing that the authors extracted transition phases and turning points, but it is completely unclear how this was done. Was that procedure sufficiently reliable? In other words, the methodology section must be significantly revised and improved.
With all of this in mind, it remains unclear how the narratives depicted in Figures were arrived at. Are they created based on the applied scheme? How? Put simply, the analytical strategy and logic underlying it are not elaborated in enough detail.
Author Response

(The authors gave the same response as above.)

Reviewer 3 Report
This is on the whole a well researched and topical essay. The essay needs to be better situated within the wider extant academic literature on radicalisation (see line 51). The definitions of radicalisation and violent extremism also need fine-tuning to better set the stage for the rest of the essay (see lines 30-35).
Minor copyedit is required to remove typos (eg. "women" in line 189).
Author Response

(The authors gave the same response as above.)

Round 2
Reviewer 1 Report
The authors have done a good job with the revised version of the manuscript. I have only two very minor editorial suggestions. First, the word "We" in the first sentence of the first full paragraph on page 8 looks it is unintentionally capitalized after the comma and seems like it should be lower-case. Second, in that same paragraph, I would suggest clarifying the sample sizes by changing "13 male and female detainees" to "13 male and 13 female detainees."
Reviewer 2 Report
The authors have responded to all of my comments satisfactorily and I have no further objections.